# Citrulline, Biomarker of Enterocyte Functional Mass and Dietary Supplement. Metabolism, Transport, and Current Evidence for Clinical Use

**DOI:** 10.3390/nu13082794

**Published:** 2021-08-15

**Authors:** Stefano Maric, Tanja Restin, Julian Louis Muff, Simone Mafalda Camargo, Laura Chiara Guglielmetti, Stefan Gerhard Holland-Cunz, Pascal Crenn, Raphael Nicolas Vuille-dit-Bille

**Affiliations:** 1School of Medicine, University of Basel, 4001 Basel, Switzerland; stefano.maric@stud.unibas.ch (S.M.); julian.muff@unibas.ch (J.L.M.); 2Institute of Physiology, University of Zurich, 8091 Zurich, Switzerland; simone.camargo@physiol.uzh.ch (S.M.C.); raphael.vuille-dit-bille@ukbb.ch (R.N.V.-d.-B.); 3Newborn Research Zurich, Department of Neonatology, University Hospital Zürich and University of Zurich, 8091 Zurich, Switzerland; 4Department of Visceral und Thoracic Surgery, Cantonal Hospital of Winterthur, 8400 Winterthur, Switzerland; laurachiara@me.com; 5Department of Pediatric Surgery, University Children’s Hospital of Basel, 4001 Basel, Switzerland; Stefan.Holland-Cunz@ukbb.ch; 6Hepato-gastroenterology and Nutrition, Hôpital Ambroise Paré, Boulogne Billancourt, APHP-Université Paris Saclay, 92100 Boulogne-Billancourt, France; pascal.crenn@aphp.fr

**Keywords:** citrulline, amino acid supplementation, L-citrulline, glutamine, arginine

## Abstract

L-Citrulline is a non-essential but still important amino acid that is released from enterocytes. Because plasma levels are reduced in case of impaired intestinal function, it has become a biomarker to monitor intestinal integrity. Moreover, oxidative stress induces protein citrullination, and antibodies against anti-citrullinated proteins are useful to monitor rheumatoid diseases. Citrullinated histones, however, may even predict a worse outcome in cancer patients. Supplementation of citrulline is better tolerated compared to arginine and might be useful to slightly improve muscle strength or protein balance. The following article shall provide an overview of L-citrulline properties and functions, as well as the current evidence for its use as a biomarker or as a therapeutic supplement.

## 1. Introduction

L-Citrulline is a non-essential and non-proteinogenic amino acid (AA), which has first been isolated by Koga out of water melon juice [1]. The presence in human proteins has been suspected for a long time [2], and it has been shown that posttranslational modification, called citrullination or deamination, plays a major role and is associated with inflammatory disease [3,4]. Citrulline has antioxidant and vasodilation properties and belongs to the human nitric oxide system [5]. In addition, both for undernourished and sarcopenic aged patients [6] and for sports purposes [7], possible anabolic effects after oral supplementation are suggested. Circulating citrulline is released from small intestinal enterocytes, predominantly in its proximal sections of jejunum and duodenum [8], where it is synthesized de novo from precursor AAs deriving from either nutritional proteins or systemic circulation [9,10]. Because the kidney is the main organ to metabolize citrulline into arginine, high plasma citrulline levels may reflect kidney failure [11]. Most nutrients do not contain relevant amounts of citrulline; however, there are about 7–14 mg citrulline per g dry weight of watermelon and 1.9 mg per g fresh weight [12]. In order to know more about the physiological properties of citrulline we performed a structured literature research and hereby summarize the signaling pathways of citrulline and the current evidence for its clinical use.

Literature research has been performed in April 2021 with the terms “citrulline” AND (“clinical trials” OR “trials”) mentioned in the abstract or title. The research databases Pubmed, EMBASe, and Cochrane Library were accessed, and case reports and animal studies have been excluded. Afterwards, doubles were erased, and the remaining literature was clustered by T.R. and J.L.M. and summarized in the following review. Important animal data that were repetitively cited in these studies retrieved above are also summarized in this review.

The structured research found 389 articles, the same search on EMBASE explicitly excluding case reports or animal studies found 301 hits, and the Cochrane Library retrieved 263 articles. A total of 184 records were at least reported twice and erased. However, 153 published articles referred to rheumatological diseases where citrullinated proteins are mainly used as biological markers for disease severity. A total of 52 articles assess the effect of citrulline supplementation on sportive executive functions. These studies show small but positive effects of citrulline supplementation on high-intensity strength that are summarized within the review by Trexler et al. [13]. A total of 44 articles deal with the potential effect of citrulline on hypertension and its effect on vessel tone. However, a meta-analysis published in 2018 analyzed the potential effect of citrulline supplementation on blood pressure but has not found any significant effect [14]. In 19 articles, citrulline as a marker for intestinal function or as a potential supplement in cases of intestinal diseases is discussed. The remaining articles either just mentioned citrulline related to certain diseases or assessed protein status and metabolics after citrulline supplementation in different persons or were unrelated to the topic. The PRISMA flow diagram is displayed in Figure 1. A summary of the included studies can be found in the Appendix A, Appendix B, Appendix C and Appendix D.

## 2. Results

### 2.1. Citrulline Precursors, Metabolism and Inter-Organ Relationship

Glutamate represents a crossroad between AA and carbohydrate metabolism. It can serve as a source of all known precursors for intestinal citrulline synthesis, which are glutamine [15,16,17], arginine [18,19], proline [20], and ornithine [9,18,21]. Glutamine has generally been considered as the main precursor of intestinal citrulline synthesis [10,17], and glutamine supplementation was shown to increase intestinal citrulline and renal arginine synthesis [17,19]. Glutamine depletion from the diet was correlated with decreased plasma citrulline levels in humans [22]. However, some studies suggest glutamine to be a nonspecific nitrogen (and carbon) donor [23]. This discrepancy results from the kind of precursors used in metabolic studies. When oral ^15^N-glutamine was used as a precursor, an enrichment in ^15^N-citrulline can be observed, and it is responsible for approximately 5% of the nitrogen of circulating citrulline [24]. When ^13^C-glutamine was used, a negligible (0.4%) incorporation into circulating citrulline was detected. Orally administered U-^13^C-arginine or U-^13^C-proline accounted for 40% and 3.4% of the circulating citrulline, respectively [20,25]. Therefore, the relative contribution of each precursor to plasma citrulline synthesis in humans remains controversial [26].

The citrulline production, metabolism, reabsorption, and turnover involve the intestines, the liver (for ureagenesis), and the kidneys, as displayed in Figure 2. Citrulline in enterocytes, as in hepatocytes, is made from ornithine in the mitochondrial matrix by the enzyme ornithine carbamyltransferase (OCT), one of the key enzymes in citrulline synthesis and in the urea cycle [21]. In contrast to hepatocytes, where the synthesized citrulline is compartmentalized as an intermediate of the urea cycle and does not contribute to systemic (circulating) citrulline flux, enterocytes show only a low expression of argininosuccinate synthetase (ASS) [27] and argininosuccinate lyase (ASL), the two enzymes that subsequentially interconvert citrulline to arginine. Citrulline, following synthesis in small intestinal enterocytes, is released across the basolateral membrane into portal circulation [10,21,28]. Unlike other AAs, citrulline is poorly taken up by hepatocytes, therefore bypassing liver metabolism and entering systemic circulation at a level of 10.4–13.6 µmol per kg and hour [21,29]. It has been shown that citrulline uptake into renal epithelial cells can occur both apically from primary urine [30] as well as basolaterally from the capillary system [30]. After filtration (at the glomeruli in the kidney) most of the citrulline is reabsorbed by proximal tubule kidney cells [31]. The almost complete reabsorption of plasma AAs prevents their urinary loss and helps to maintain homeostasis [32]. Proximal kidney tubule cells hereby use a similar set of luminal and basolateral amino acid transporters (AATs) for AA reabsorption from the primary urine as the small intestine for absorption of digested dietary proteins [32]. In the proximal kidney tubule cells, citrulline is converted by ASS and ASL into arginine, which is released into systemic circulation for use by peripheral tissues. Citrulline delivery to the kidney (and therefore circulating plasma citrulline concentration) represents the rate-limiting step of renal arginine synthesis [33]. Pharmacokinetic studies with oral citrulline supplementation have shown a dose-dependent increase in plasma citrulline, arginine, and ornithine levels [34]. Furthermore, plasma citrulline has been shown to be augmented following oral citrulline supplementation [22].

This metabolic interaction between the small intestine and the kidneys is known as the intestinal-renal axis and is believed to provide arginine supply to peripheral tissues, which would otherwise be taken up by the liver and induce ureagenesis and hence AA catabolism [35]. Citrulline is therefore seen as a form to avoid excessive hepatic metabolism of AAs. This mechanism is mainly activated in conditions with low protein intake, as in a post-absorptive (fasting) state [36]. However, in very preterm children, citrulline might be converted into arginine directly by the gut in situ [37]. Citrulline may serve to support protein anabolism in states of low protein intake [38,39]; this way, citrulline helps to limit the plasmatic arginine decrease [40,41]. Moreover, endogenous arginine and citrulline production is increased in the case of lowgrade inflammation with increased NO production rates such as chronic obstructive lung disease [42]. The intestinal-renal axis underlies a maturation process because kidneys produce arginine from citrulline in the presence of ASS and ASL, which is differentially expressed depending on age [43]. Consequently, plasma citrulline concentration is lower in neonates compared to adult individuals and increases during development [21,44]. If the kidneys are dysfunctional, citrulline is directly metabolized to arginine in the enterocytes, which show a high expression of ASS and ASL, but low arginase expression [15,36].

### 2.2. Causes for High and Low Plasma Citrulline Levels

Normal plasma citrulline concentrations in healthy adults have been defined as 40 (±10) µmol/L [45,46]. Both elevations and reductions in citrulline levels can either be inherited or acquired and are summarized in Table 1. Increased citrulline levels can be caused by rare inborn errors of disease such as citrullinemia, which is caused by a deficiency of ASS that leads to elevated levels of blood citrulline and ammonia, ending in hyperammonemic coma and early death [47,48]. Likewise, the deficiency of ASL accumulates argininosuccinic acid with deficient endogenous arginine production and high levels of ammonia and neurocognitive decline [49]. If ASS and ASL activity are reduced due to kidney failure, citrulline elevations may be noticed, too [50]. The adult-onset type II citrullinemia is caused by a defect in the mitochondrial aspartate-glutamate carrier [51]. Unlike the disease conditions described above, a physiological increase in citrulline has been described after improvement of the intestinal absorption capacity, such as after intestinal lengthening [52], bariatric surgery [53], or after enterotrophic treatment with teduglutide (glucagon-like peptide 2), which increases intestinal mucosal growth and trophic function [54,55]. In contrast, lowered citrulline levels have been detected due to deficiencies in carbamoyl phosphate synthetase 1 (CPS1) [56] and OTC [57] because they limit the turnover from ornithine to citrulline in mitochondria. It seems that prematurity can potentially be associated with lowered citrulline levels [58], but it is difficult to differentiate this lowering from lower levels due to any inflammatory process. Any condition that is associated with a reduced absorptive intestinal capacity has been associated with reduced citrulline levels and will be discussed in detail (see below).

The importance of intestinal nutrient absorption becomes evident in situations of intestinal failure. Intestinal failure reflects the reduction in functional small bowel below the minimum necessary for digestion and absorption to maintain growth in children and/or homeostasis in children and adults. The most common etiology is short bowel syndrome (SBS), which describes a reduction in anatomical and functional bowel length [64,65,66,67]. Several diseases of the gastrointestinal tract, including necrotizing enterocolitis (NEC), intestinal atresia [68], midgut volvulus, and long-segment Hirschsprungs’ disease in children, as well as mesenteric ischemia, Crohn’s disease, and irradiation in adults [65], may lead to extensive damage and/or intestinal resections ending in SBS [69]. Patients with chronic intestinal failure can be dependent on long-term parenteral nutrition with its inherent morbidity and mortality, including (repeated) catheter-associated sepsis, cirrhosis, and liver failure [69]. While many factors seem to play a role, the length of the remnant intestine and the type of digestive anastomosis reflects a major determinant of patient survival and nutritional prognosis [67,70,71].

Significant reduction in plasma citrulline concentration has been shown in various pathologies of the digestive tract [46], including NEC [72], SBS [73,74,75,76,77,78], villous atrophy including celiac disease [8], acute mucosal enteropathy of various etiology such as mucositis after antineoplastic treatment, chemotherapy and/or radiotherapy [79,80], HIV enteropathy [81] and acute enteric infection or graft rejection after short bowel transplantation [82,83]. Plasma citrulline can also be decreased in critically ill patients with intestinal dysfunction in the intensive care unit [63,84]. Interestingly, early antibiotic use seems to be associated with lower citrulline levels and lower performance and survival rates in patients with non-small-cell lung cancer, which might possibly be associated with the changed microbial profile [85]. In post-surgical conditions (SBS) the threshold, for the parenteral nutrition autonomy is 20 mol/L, whereas, in medical conditions, the threshold is 10 mol/L [8,45,46,86] **.** Some of these studies also suggested a correlation between plasma citrulline and intestinal adaptation after small bowel resection, the dependence of nutritional support, and absorptive function of the intestine [73,74,75,76]. Finally, plasma citrulline was elevated in animal models and pediatric patients following intestinal lengthening using serial transverse enteroplasty (STEP) [87,88]. Consequently, citrulline is a potential sensitive biomarker for small intestinal absorptive function [45,86,89]. It can be clinically useful to monitor citrulline levels of patients before and after intestinal surgery and to predict absorption even before enteral feeds are started [61].

Surgical treatments aim to elongate the small intestine in order to increase its absorptive capacity, especially in pediatric patients. Among different surgical treatment options, longitudinal intestinal lengthening and tailoring (LILT), first described by Bianchi in 1980 [90], and STEP are the most commonly used [87]. Unfortunately, both procedures reconfigure the intestinal morphology (making a long thin tube out of a short thick tube) without creating more luminal surface area. As the LILT procedure is technically very demanding and more prone to complications, the outcomes following STEP seem to be more favorable and can potentially be repeated [88].

### 2.3. Citrulline and Cancer

In nine children with AML, citrulline was significantly lower after chemotherapy (27 plasma samples) and positively correlated with scores for mucosal integrity [91]. Hepatocellular carcinoma, which often lacks ASS, is commonly dependent on arginine metabolism. Consequently, several studies assessed arginine-depleting strategies such as treatment via pegylated arginine deiminase [92] or arginase [93]. Both treatment strategies led to higher plasma levels of citrulline. There is a debate whether arginine depletion might also be useful in other ASS deficient cancer types. In a study including 68 patients with ASS1 deficient mesothelioma, Szlosarek demonstrated a prolonged progression-free survival and a reciprocal increase in citrulline after treatment with pegylated arginine deiminase [94]. When this medicament was used against glioma-, melanoma- or other ASS1 deficient malignancies, a similar rise of citrulline has been noted [95,96]. Likewise, parenteral glutamine substitution has been associated with higher citrulline levels [97]. However, in melanoma patients, a decrease in citrulline levels after treatment with pegylated interferone y was noted, which has been attributed to lower NO production [98]. The largest study of citrulline assessment in cancer patients involves 957 patients where citrullinated histone, an accepted marker for neutrophil extracellular trap (NET) formation, was shown to correlate with patient mortality [99].

### 2.4. Intestinal Amino Acid Transporters and Transport Mechanisms—Application to Citrulline

AAs are polar molecules and therefore rely on a variety of transport proteins to cross the lipid bilayer of cell membranes (such as small intestinal enterocytes or proximal kidney tubule cells). AATs are a heterogeneous group of transmembrane proteins that vary in terms of substrate specificity, transport mechanism, transport kinetics, tissue-specific expression, cellular distribution within a cell, and dependence on protein subunits [100]. In this review, we will focus on apical membrane transporters involved in the uptake of L-citrulline and especially its precursors. At the basolateral membrane, we will discuss transporters involved in citrulline efflux from cells. More information about AAT in epithelial cells (not restricted to citrulline and its precursors) can be found elsewhere [100,101,102,103].

### 2.5. Brush-Border Membrane Transporters for Citrulline Precursors in Small Intestinal Enterocytes

Transport of almost all neutral AA across the apical enterocyte (and proximal tubule) membrane is largely mediated by the transporter B^0^AT1 (SLC6A19), a broad specificity sodium-dependent symporter using the sodium gradient created by the basolateral Na-K-ATPase as driving force [104,105,106]. Heterodimeric AAT b^0,+^AT-rBAT (SLC7A9-SLC3A1) functions as an obligatory AA exchanger providing transport (exchange) of cationic (such as arginine) and neutral AAs [107]. b^0,+^AT-rBAT is composed of two subunits, a type II membrane protein (heavy chain; b^0,+^AT) and a polytopic membrane protein (light chain; rBAT) bound together via a disulfide bridge [108,109]. Other luminal membrane transporters such as IMINO transporter SIT1 (SLC6A20) and the proton-dependent AA transporter PAT1 (SLC36A1) provide transport of proline, glycine, and some other neutral AAs to some extent [32]. Interestingly, both neutral AATs, B^0^AT1 and SIT1 (but not the other transporters mentioned) depend on the presence of accessory proteins for being expressed in the luminal cellular membrane, whereas the B^0^AT1 and SIT1 expression in kidney proximal tubule cells’ brush-border membrane depends on co-expression of type I membrane protein collectrin (TMEM27) [110], expression of the same transporters in small intestinal enterocytes depends on the presence of its structural homolog ACE2 (as shown in ACE2 knock-out mice) [32,111]; ACE2 (angiotensin-converting enzyme 2) is a membrane-bound monocarboxypeptidase that hydrolyzes luminal peptides and provides AAs for transmembrane transport. Moreover, ACE2 has also been identified as a functional receptor for the SARS coronavirus (SARS-CoV) in 2003 and more recently for SARS-CoV-2 [68,112,113,114,115]. Finally, the proton-dependent peptide transporter PEPT1 (SLC15A1) transports citrulline precursors (including glutamine, arginine, glutamate, and proline) as di- or tripeptides [116]. PEPT1 seems to be important to provide sufficient AA uptake when AATs become saturated after high dietary protein intake [117].

### 2.6. Citrulline Transport—Luminal Membrane

As previously indicated, citrulline is a non-proteinogenic AA. Therefore, it is not incorporated in human protein biosynthesis, nor does it appear in nutritional proteins in substantial quantities, except essentially in watermelon [12]. Citrulline transport across the luminal enterocyte membrane hence seems to be of minor importance. As a functional characterization of AA transporters (i.e., testing substrate specificities of different AAs for a certain transporter) mainly focused on proteinogenic AA, transport of citrulline has not been shown yet and remains unknown (to the best of our knowledge). Based on transported substrates and their structural similarity with citrulline, it has been suggested that citrulline transport across the luminal membrane of proximal tubule kidney cells (and small intestinal enterocytes) is mediated by AATs B^0^AT1 (SLC6A19) and b^0,+^AT (SLC7A9) [31,118,119].

### 2.7. Amino Acid Transport—Basolateral Membrane

Basolateral AA efflux from small intestinal enterocytes is mediated by two different types of transporters: uniporters and heterodimeric AA exchangers. To enable net efflux of all proteinogenic AAs from small intestinal enterocytes, a functional interaction between these different transport types is necessary. Equilibration of essential neutral AAs along their concentration gradients between enterocytes and the extracellular space is mediated by the two low-affinity uniporters for essential AAs, LAT4 (SLC43A2) and TAT1 (SLC16A10). LAT4 belongs to the sodium-independent large neutral AA transporter family “system L” and acts as low-affinity facilitated diffusion protein for branched-chain AAs (leucine, isoleucine, valine) [120], as well as for phenylalanine and methionine [121]. TAT1 transports aromatic AAs (tyrosine, tryptophan, phenylalanine) [122,123].

Neutral and cationic AAs are transported by the exchangers LAT1 (SLC7A5), LAT2 (SLC7A8), y^+^LAT1 (SLC7A7), and y^+^LAT2 (SLC7A6), with LAT2 and y^+^LAT1 being expressed primarily and at a much higher level in resorbing epithelia such as the small intestine and the renal tubular cells [124,125]. The heavy chain 4F2 (SLC3A2) hereby binds to different light chains in the basolateral membrane, including LAT1, LAT2, y^+^LAT1, and y^+^LAT2 [108,109].

### 2.8. Citrulline Transport—Basolateral Membrane

As for B^0^AT1 and b^0,+^AT in the luminal membrane, transport of citrulline across the basolateral membrane remains unknown from the literature, as mainly proteinogenic AAs were tested as putative transporter candidates in the past (to the best of our knowledge). Reviewing transport specificities and structural similarities of proteinogenic AAs (to citrulline) accepted by the named transporters makes LAT2 and y^+^LAT1 the most valuable candidates for basolateral citrulline transport [109,119,126]. As these AA exchangers exchange AAs in a 1:1 ratio, they rely on the co-expression of uniporters such as LAT4 and TAT1. As LAT4 has a quite narrow substrate specificity [120,121], citrulline transport by LAT4 seems unlikely. Nevertheless, citrulline levels in amniotic fluid and plasma were reduced in LAT4 knock-out mice, indicating LAT4 as a functional partner providing extracellular substrates for LAT2/y^+^LAT1-mediated citrulline exchange [28,121].

### 2.9. Citrulline and Intensive Care Treatment

The review found 11 articles that referred to ICU treatment; however, most of them deal with arginine or protein supplementation. In six studies, citrulline is men-tioned in detail: in these patients. Three studies measured serum citrulline in these patients. Ware et al. analyzed 135 patients with severe sepsis and found out that those 44 with ARDS at ICU entrance had lower citrulline levels (mean 6, IQR 3.3–10.4 µmol/L) compared to those without ARDS (mean 10.1 (6.2–16.6 µmol/L)). Piton analyzed the effect of 3 days of enteral nutrition in patients with severe sepsis and found that those with enteral nutrition rose from 12.2 to 18.7 µmol/L while those on parenteral nutrition who started with 13.3 had a citrulline level of 15.3 µmol/L. In patients with severe sepsis, only low levels of citrulline were found, but no significant difference between survivors and non-survivors. Groups differed concerning their glutamine level, which was higher in those who died.

### 2.10. Citrulline in Intestinal Development

Preterm infants are often dependent on parenteral nutrition at least for one week until their enteral feds are established. Bourdon found only weak correlations with post-conceptional age, parenteral amino acid supply, and daily volume of enteral mixture administered. They found that urinary citrulline cannot predict GI tolerance, but the major determinant of urinary citrulline may be arginine produced by NO-synthase [37].

### 2.11. Citrulline and Intestinal Microbiota

Citrulline changes are associated with microbial changes in the gut, for example, after chemotherapy [127]. Gut bacteria are known to use AAs, including arginine, for both protein synthesis and the production of arginine-derived metabolites such as polyamines or nitric oxide, as reviewed by Baier et al. [128]. Interestingly, Indian women, especially those with very light babies, had lower citrulline and arginine flux compared with Jamaican or American women, which has been associated with microbial dysbiosis [129]. In addition, the bacterium *Porphyromonas gingivalis,* which is associated with the development of periodontitis, is also able to convert arginine to citrulline with the aid of the peptidylarginine deiminase, a process called “citrullination” [130]. This protein, which contains the AA citrulline, is recognized by the anti-citrullinated protein antibodies, which are highly specific for rheumatoid arthritis [131,132].

### 2.12. Clinical and Therapeutical Implications

With its diverse biological functions, citrulline suggests several therapeutic applications, as summarized in Table 2. As a precursor of arginine [126], it might be useful when the arginine turnover is high such as during hemolysis or liver damage. As a precursor of nitric oxide (NO), it might support the treatment of pulmonary hypertension [133]. Improved arginine recycling by citrulline [134] might improve T cell function [135].

### 2.13. Safety of Oral Citrulline Supplementation

Whereas some studies indicate gastrointestinal side effects from oral L-arginine supplementation, including nausea, abdominal cramping, and diarrhea [149], no side effects were seen when oral citrulline was administered [22,116]. Indeed, no toxicity was identified when oral citrulline had been administered to infants and children in doses up to 3.8 g/m^2^ per day (in five doses of 1.9 g/m^2^ every 12 h) [116] and in doses up to 15 g in healthy human adults [34]. Due to its limited degradation in the placenta [150], it seems to be a promising supplement for pregnant women. Animal data suggest that it might be beneficial for the prevention of intrauterine growth restriction [151,152]. In 24 obese pregnant women, citrulline has been used at a dose of 3 g/day for 3 weeks, which has been associated with improved vascular function and blood pressure without any side effects [153].

Furthermore, long-term citrulline administration in patients with urea cycle defects was without any side effects [154]. Finally, citrulline administration via the intravenous route has been performed in infants and young children without any side effects (including severe systemic hypotension) [155].

### 2.14. Oral AA Supplements to Induce Nitric Oxide-Mediated Vasodilation

Nitric oxide (NO) is a vasoactive gaseous signaling molecule that induces vasodilation in both arterial and venous blood vessels [156]. In endothelial cells, NO is synthesized from arginine by eNOS (endothelial-nitric oxide synthase). Reduced eNOS synthesis associated with aging contributes to endothelial dysfunction. Decreased NO bioavailability impairs blood flow and increases the risk of hypertension, atherosclerosis, insulin resistance, and cardiovascular disease [133]. As arginine and its precursor citrulline are intermediates in the urea cycle and substrates for nitric oxide production, their supplementation has been investigated, at various doses, in the treatment of endothelial dysfunction and related diseases (including arterial hypertension, pulmonary arterial hypertension, pressure sores, erectile dysfunction, arteriosclerosis, some mitochondrial disorders, and necrotizing enterocolitis) [116,157,158]. Arginine has a relatively high first-pass extraction in the intestine and the liver (as arginases 1 and 2 metabolize arginine to ornithine and urea). Furthermore, oral arginine supplementation may cause (dose-dependent) gastrointestinal distress (unlike citrulline), resulting in higher activity and bioavailability of citrulline as compared to arginine [133]. Different animal studies have shown a protective effect of dietary citrulline supplementation by preserving eNOS synthesis and NO production against an atherogenic diet [159,160]. Furthermore, oral citrulline shows antioxidant effects by reducing reactive oxygen species (ROS) (NO-dependent and NO-independent), thereby preventing platelet aggregation and pathological vascular remodeling [133]. By increasing endogenous arginine and hence NO synthesis, citrulline was shown to reduce arterial stiffness and also had anti-hypertensive effects: oral citrulline or watermelon extract supplementation for a few weeks resulted in blood pressure reductions in pre-hypertensive and hypertensive patients [161]. It is to note that a recent meta-analysis did not find any significant beneficial effect of citrulline on arterial, systolic or diastolic, blood pressure [14], but this area needs further rigorous clinical trials. In a randomized controlled trial of 40 children randomized to five perioperative doses (1.9 g/m^2^/dose), oral citrulline or placebo found that either children with naturally elevated citrulline or citrulline due to supplementation did not develop postoperative pulmonary hypertension [162]. Moreover, in obese asthmatics with low or normal fractional excretion of NO L-citrulline treatment (15 g/d for 2 weeks) improved asthma control [163].

### 2.15. Oral L-Citrulline Supplementation to Improve Exercise Performance in Healthy Athletes

Oral citrulline supplementation has been shown to increase pulmonary oxygen uptake and exercise performance in healthy human probands and athletes [7,164]. Nevertheless, results from different studies were not uniform as others could not show an effect on exercise performance upon oral citrulline supplementation [165]. Furthermore, oral citrulline was supplemented as a malate salt, possibly biasing obtained results. It hence remains unclear if citrulline itself or the Krebs cycle intermediate malate improved exercise performance.

### 2.16. Citrulline Supplementation in Children

Citrulline supplementation has shown to be safe in children, the group of Marealle and Cox used 0.1 mg/kg/day in ready to use supplementary food (RUSF) [166,167], while Silvera Ruiz calculated 3 g/m^2^/day [154] for long term supplementation (4 months). Citrulline can be a marker for intestinal function which is reduced in malnourished children [168], in those with necrotizing enterocolitis [169] or in case of severe mucositis [170]. Citrulline may increase with gluten free diet in children with celiac disease [171]. However, it is not a marker of gastrointestinal tolerance [37] and has shown to be higher in a group of children formerly born preterm compared to their term counterparts [172], while shortly after delivery preterm citrulline levels have shown to be very low [173]. As it improves the supply with NO, it has been used to lower pulmonary hypertension [162,174]. As in adults, severely ill children have lower amounts of serum citrulline than healthy children [175]. A more detailed overview is shown in Appendix A [162]

### 2.17. Citrulline Supplementation and Exercise Performance in Sarcopenic Elderly Patients

Sarcopenia refers classically to the loss of skeletal muscle mass, power, and strength due to aging and/or immobility [176]. Sarcopenia may lead to disability and reduced quality of life and is considered to be part of frailty syndrome, which refers to the progressing decline in health and function typically occurring in geriatric patients [177]. Moreover, sarcopenia is possible in all chronic conditions such as inflammatory diseases, chronic liver and intestinal diseases, undernutrition of various causes, and cancers [178]. In addition to loss of muscle mass and strength, mitochondrial oxidative capacity likewise deteriorates as humans age, resulting in reduced exercise performance [179,180]. Oral citrulline supplementation has shown anabolic effects on muscle protein synthesis in malnourished animals [181] as well as higher systemic AA availability, but has no significant effect, with an oral dose of 10 g per day during 3 weeks, on protein synthesis in sarcopenic malnourished patients of more than 80 years [6]. However, in this last study, citrulline supplementation was associated with a higher systemic AA availability, and in the subgroup of women, citrulline supplementation increased lean mass and appendicular skeletal muscle mass and decreased fat mass. Inconsistent findings were also found when citrulline effects on protein synthesis in healthy humans were assessed [39,182]. In nine adult SBS patients in suitable nutritional status, in the late phase of intestinal adaptation and with near-normal baseline citrulline homeostasis, oral citrulline supplementation (0.18 g/kg/d during 7 days) enhanced citrulline and arginine bioavailability but did not have any anabolic effect on whole-body protein metabolism [183]. Whether oral citrulline would impact whole-body protein anabolism in severely malnourished SBS patients in the early adaptive period, and with baseline plasma citrulline below 20 μmol/L, is not known. In addition, the mechanism of citrulline action on muscle protein synthesis (anabolic effect but not anti-catabolic) is hereby not completely understood and may involve the mTOR (mammalian/mechanistic target of rapamycin) pathway, iNOS, insulin secretion, and vasodilation effects [5,133] and/or reallocating ATP consumption [143]. Major surgical procedures (tumor resections, etc.) are frequently necessary for the elderly and often lead to a further decline in frailty in these patients. As physical fitness, mood, and nutritional status have been shown to affect outcome following major surgical procedures, especially in sarcopenic and/or frail patients, not only the post- but also the presurgical period has been recognized as an important time span to improve exercise tolerance, optimize the nutritional status, as well as psychological wellbeing of the patients, which is referred to as multimodal prehabilitation [184,185]. Evidence is currently not sufficient to recommend citrulline supplementation in frail patients, but possible anabolic effects warrant further investigation.

### 2.18. Oral Citrulline Supplementation to Improve Non-Alcoholic Fatty Liver Disease

Another situation of the potential interest of citrulline, due to its anti-inflammatory and antioxidant actions with reduction in hypertriglyceridemia and liver fat accumulation induced by diet (mainly fructose), is metabolic liver disease (NAFLD, i.e., steatosis), but at the present time, after animal preclinical data in rats with fructose-induced non-alcoholic liver disease [146,148], there is only one clinical promising study with a low dose of 2 g/d during 3 months [186].

## 3. Conclusions

Citrulline appears to be a suitable functional biomarker for severe intestinal disease, no matter whether the intestinal dysfunction developed because of surgical, chemotherapeutical, or radiological intervention or due to a medical condition. It can be used as a marker for the intestinal function as a follow-up under adapted care such as nutritional care, surgery, or pharmacological treatments. Citrulline is a potential therapeutic tool, which may be used as a dietary supplement and as a NO donor bypassing the metabolism of arginine. The optimal dose, application route (either intravenous or oral), and timing for citrulline supplementation need further investigation.

## Figures and Tables

**Figure 1 nutrients-13-02794-f001:**
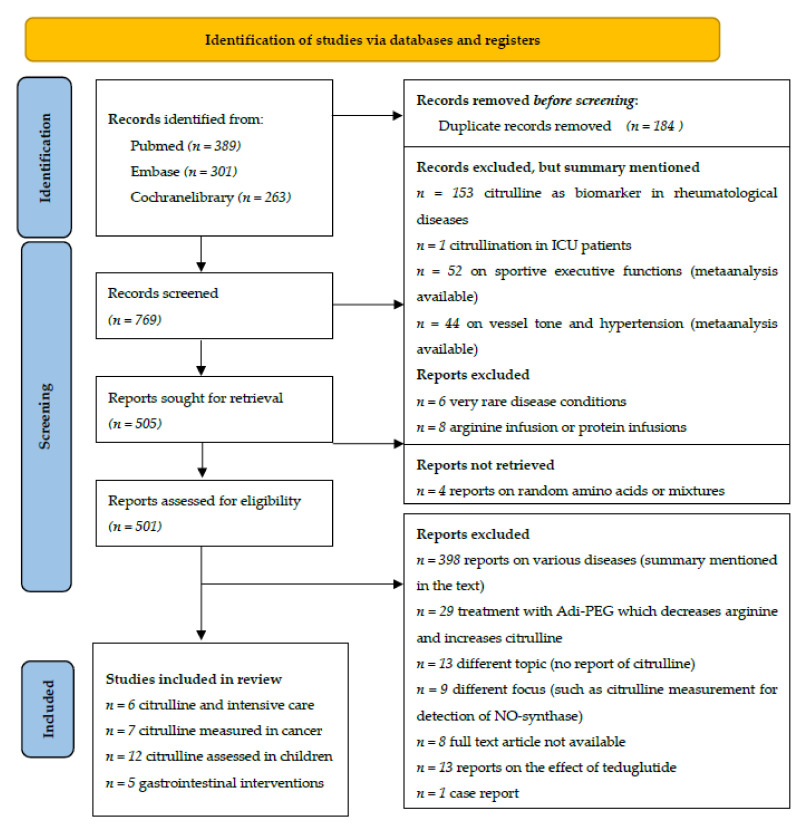
Literature research adapted from [15], table licensed under creative commons.

**Figure 2 nutrients-13-02794-f002:**
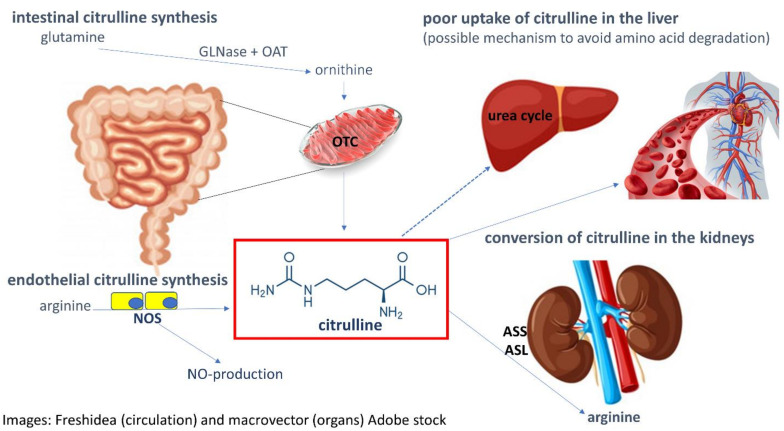
This image demonstrates how citrulline is synthesized, converted, and degraded. Abbreviations: ASL argininosuccinate lyase, ASS argininosuccinate synthetase, GLNase glutaminase, OAT ornithine aminotransferase, OTC ornithine trans-carbamylase, NOS nitric oxygen synthase.

**Table 1 nutrients-13-02794-t001:** Main conditions that induce or decrease plasma citrulline concentration.

Elevated Citrulline	Decreased Citrulline
Rare metabolic deficiencies: argininosuccinate synthase (ASS) [47,48]; argininosuccinic acid lyase (ASL) [49]; mitochondrial aspartate-glutamate carrier [51] Intestinal lengthening [54] Enterotrophic treatment with teduglutide (glucagon-like peptide 2) [37] Bariatric surgery [36] Renal insufficiency [11]	Rare metabolic deficiencies: carbamoyl phosphate synthetase 1 (CPS1) [56]; Ornithine transcarbamylase (OTC) 57 Prematurity/before weaning [58] Mucositis due to chemo- or radiotherapy [59,60] Short bowel/gut syndrome (intestinal failure) [61]
	Villous atrophy: celiac disease, various intestinal diseases [45]
	Graft rejection after small bowel transplantation [62]
	Intestinal dysfunction in intensive care conditions [63]

**Table 2 nutrients-13-02794-t002:** Physiological function and potential medical use of citrulline as suggested by animal experiments.

Physiological Function	Potential Medical Use
Precursor of arginine [136]	Counteracts arginine deficiency such as during conditions of increased arginase activity (hemolysis, liver damage) [137] Protects against cerebral malaria [138]
Precursor of nitric oxide (NO) [139]	Reduces blood pressures in hypertension [133] Vasodilator for pulmonary hypertension [140,141] Improvement of erectile dysfunction [142]
Improvement of arginine recycling [134]	Improved T cell function [135]
Increased protein synthesis [143]	Counteracts sarcopenia state [6,144]
Hydroxyl radical scavenger [145] Reduction in LPS-induced inflammation [146]	improves the capacity of neuronal networks during aging [147] Attenuates fructose-induced non-alcoholic fatty liver disease [148]

## Data Availability

No public data storage available.

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
