# Peer review of "Citrulline, Biomarker of Enterocyte Functional Mass and Dietary Supplement. Metabolism, Transport, and Current Evidence for Clinical Use"

_nutrients, 2021, doi:10.3390/nu13082794_

Round 1
Reviewer 1 Report
July 22nd, 2021
Dear Authors.
I was very happy to review your manuscript “Citrulline, biomarker of enterocyte functional mass and dietary supplement. Metabolism, transport and current evidence for clinical use”. I reviewed this paper and will accept it after minor revision to publish in Journal “Nutrients”. L-Citrulline is a non-essential but still important amino acid released by enterocytes and you have well summarized the effects of L-Citrulline. In this manuscript, you have provided an overview of the properties and functions of L-citrulline and the current evidence for its use as a biomarker or therapeutic adjuvant. I believe this manuscript will provide useful information for researchers on the effects of L-Citrulline.
Thank you,
Sincerely yours
Reviewer.
Minor Comments:
- Please add the amount of citrulline in watermelon for fresh weight by referring to the literature below.
- Rimando, A.M.; Perkins-Veazie, P.M. Determination of citrulline in watermelon rind. J. Chromatogr. A 2005, 1078, 196–200.
- Line 44, Please delete the number 2.
- Line 134, Please delete the curly bracket.
Author Response
Dear Editors, dear reviewers
Thank you very much for the sound reviewing process. We are sorry for our delayed response which was due to the fact, that both my laptops were stolen out of my car and that I additionally had limited internet access without any mobile device.
However I tried to reconstruct the revisions, but I have not been able to reformat the whole citations, which I will do as soon as I get a new computer with a new citation program, which should be fesible around August 9th.
Thank you for your patience,
Tanja Restin
Reviewer 1:
- Please add the amount of citrulline in watermelon for fresh weight by referring to the literature below.
We added this information in line 41: Most nutrients do not contain relevant amounts of citrulline, however there are about 7-14 mg citrulline per g dry weight of watermelon and 1.9 mg per g fresh weight [12]
- Line 44, Please delete the number 2. We deleted this number
- Line 134, Please delete the curly bracket We deleted this bracket

Reviewer 2 Report
I was impressed by the fact that the articles on citrulline were fairly extracted and summarized, mentioning not only the good points but also the side effects.
The points I would like to see improved are as follows
1. The abbreviations were written only in the text, making it difficult to read. I thought it would be easier to understand if they were also written in the figures and tables.
2. I found it difficult to understand the figure 2, especially the citrulline metabolism in the kidney. I think it is difficult to express the differences in metabolism between the intestines, kidneys, and liver, the movement of metabolic products in the body, and where nitric oxide is produced in the body as an image diagram, but could you make it a little easier to understand? For example, you could remove the realistic illustrations of the organs and just use a line to separate the organs from each other.
3. I thought it would be good if the safety for pregnant women was also described in 3.13.
Author Response
Dear Editors, dear reviewers
Thank you very much for the sound reviewing process. We are sorry for our delayed response which was due to the fact, that both my laptops were stolen out of my car and that I additionally had limited internet access without any mobile device.
However I tried to reconstruct the revisions, but I have not been able to reformat the whole citations, which I will do as soon as I get a new computer with a new citation program, which should be feasible around August 9th.
Thank you for your patience,
Tanja Restin
Reviewer 2:
- The abbreviations were written only in the text, making it difficult to read. I thought it would be easier to understand if they were also written in the figures and tables.
We added the abbreviations within the figures and tables
- I found it difficult to understand the figure 2, especially the citrulline metabolism in the kidney. I think it is difficult to express the differences in metabolism between the intestines, kidneys, and liver, the movement of metabolic products in the body, and where nitric oxide is produced in the body as an image diagram, but could you make it a little easier to understand? For example, you could remove the realistic illustrations of the organs and just use a line to separate the organs from each other.
We now tried to adapt the figure and structured it into synthesis, conversion and degradation and hope it is now clarified.
- I thought it would be good if the safety for pregnant women was also described in 3.13
Thank you, we adapted this section accordingly in line 381ff: “Due to its limited degradation in the placenta (195) it sems to be a promising supplement for pregnant women. Animal data suggest that it might be beneficial for prevention of intrauterine growth restriction (196, 197). In 24 obese pregnant women citrulline has been used at a dose of 3g/day for 3 weeks which has been associated with improved vascular function and blood pressure without any side effects (194)”.
- Powers, R. et al., L-citrulline administration increases the arginine /ADMA ratio, decxrases blood pressure and improves vascular function in obese pregnant women, Prenancy Hypertension: An International Journal of Women’s Cardiovascular health, Volume 5, Isseue 1, 2015, p. 4
- Lassala, A.; Bazer, F.W.; Cudd, T.A.; Li, P.; Li, X.; Satterfield, M.C. Intravenous Administration of L-
Citrulline to Pregnant
Ewes Is More Effective Than L-Arginine for Increasing Arginine Availability in the Fetus. J. Nutr. 2009,
139, 660–665
- Tran, N.; Amarger, V.; Bourdon, A.; Misbert, E.; Grit, I.; Winer, N.; Darmaun, D. Maternal citrulline ¨
supplementation enhances
placental function and fetal growth in a rat model of IUGR: Involvement of insulin-like growth factor 2
and angiogenic factors. J. Matern. Fetal Neonatal Med. 2017, 30, 1906–1911.
- Bourdon, A.; Parnet, P.; Nowak, C.; Tran, N.; Winer, N.; Darmaun, D. L-Citrulline Supplementation
Enhances Fetal Growth and Protein Synthesis in Rats with Intrauterine Growth Restriction. J. Nutr.
2016, 146, 532–541